# Identification and Targeting of Mutant Peptide Neoantigens in Cancer Immunotherapy

**DOI:** 10.3390/cancers13164245

**Published:** 2021-08-23

**Authors:** Daniel J. Verdon, Misty R. Jenkins

**Affiliations:** 1Immunology Division, The Walter and Eliza Hall Institute of Medical Research, Parkville, VIC 3052, Australia; jenkins.m@wehi.edu.au; 2Department of Medical Biology, The University of Melbourne, Parkville, VIC 3052, Australia; 3La Trobe Institute of Molecular Science, La Trobe University, Bundoora, VIC 3086, Australia

**Keywords:** neoantigen, public neoantigen, private neoantigen, adoptive cell transfer, immunotherapy, tumour-infiltrating lymphocytes, T cells, TCR, cancer, checkpoint blockade, melanoma, mass spectrometry, whole exome sequencing, vaccination

## Abstract

**Simple Summary:**

Cancerous cells acquire genetic mutations that can lead to changes in the amino acid sequence of proteins. These altered amino acid sequences, or “neoantigens” allow the immune system to recognize the mutated cells as “non-self” and eliminate them. This review outlines discoveries that identified neoantigens as a key immune target. Further, we discuss the development of bioinformatic and DNA sequencing technologies used to detect patient-specific mutations giving rise to neoantigens, and the methods by which neoantigens can be targeted in cancer therapy.

**Abstract:**

In recent decades, adoptive cell transfer and checkpoint blockade therapies have revolutionized immunotherapeutic approaches to cancer treatment. Advances in whole exome/genome sequencing and bioinformatic detection of tumour-specific genetic variations and the amino acid sequence alterations they induce have revealed that T cell mediated anti-tumour immunity is substantially directed at mutated peptide sequences, and the identification and therapeutic targeting of patient-specific mutated peptide antigens now represents an exciting and rapidly progressing frontier of personalized medicine in the treatment of cancer. This review outlines the historical identification and validation of mutated peptide neoantigens as a target of the immune system, and the technical development of bioinformatic and experimental strategies for detecting, confirming and prioritizing both patient-specific or “private” and frequently occurring, shared “public” neoantigenic targets. Further, we examine the range of therapeutic modalities that have demonstrated preclinical and clinical anti-tumour efficacy through specifically targeting neoantigens, including adoptive T cell transfer, checkpoint blockade and neoantigen vaccination.

## 1. Introduction

The T cell arm of the adaptive immune system is exquisitely sensitive at discriminating “self” from “non-self”. It achieves this both through stringent deletion (through negative selection) of cells expressing potentially self-peptide-reactive TCR in the thymus, and through the requirement for naïve T cells to recognize their cognate peptide epitope presented in the context of MHC-I/II by antigen-presenting cells (APCs). APCs mature via exposure to pathogen- or cellular stress- related inflammatory signals prior to secondary lymphoid organ trafficking and are thus capable of providing both soluble and cell-contact-mediated co-stimulatory signals. For a tumour to be recognized as “non-self” and induce a T cell mediated response, it must first express altered peptide sequences and interact with APCs in an inflammatory context sufficient to induce maturation and naïve T cell priming. Subsequently, altered peptide sequences are then presented in the context of endogenous MHC molecules to allow subsequent recognition by expanded T cell effectors.

Tumour cells can acquire expression of peptides recognized as “non-self” in several ways. Many viral infections are oncogenic—prominent examples include EBV-induced lymphomas and HPV16-induced oropharyngeal and cervical carcinomas. Peptides derived from viral proteins can be processed and presented on transformed cells as they would during any viral infection. Such peptides are highly immunogenic and have been successfully targeted therapeutically [1], but are not “altered-self”-sequences and are not further discussed in this review.

Tumour-associated or lineage-differentiation antigens are self-proteins expressed in a tightly lineage-restricted pattern and conserved in cancers of that tissue of origin—one well-characterized suite being the melanoma differentiation antigens (MDA) involved in melanosome biogenesis: MART-1/Melan-A, Tyrosinase, gp100/pmel and Tyrosinase-relaed-protein-1 and -2 [2,3,4]. Despite being “self”, these proteins exhibit incomplete central tolerance and reactive TCR can escape negative selection by virtue of truncated protein expression (MART-1/Melan-A) or transcriptional silencing (TRP-2) in medullary thymic epithelial cells (MTEC) [5,6,7,8]. Although these antigens have been targeted clinically, typically by infusion of clonal or Transgenic (Tg)TCR products specific for MART-1/Melan-A derived peptides presented in a HLA-A*02:01 context, this is associated with severe on-target off-tumour toxicity against healthy melanocytes in the skin, eye and gut, limiting their utility as therapeutic targets [9,10,11].

Transformed cells frequently exhibit global hypomethylation, enabling the re-expression of “cancer/germline” antigens (CGA) expressed during gametogenesis and then epigenetically silenced in adult tissues. Again, these antigens are fundamentally “self”, but are not expressed in MTEC and as such viewed as entirely foreign by the immune system [12]. The best described CGA are NY-ESO-1 [13] and the extensive MAGE family [14] in melanoma, although over 100 CGA have been described. NY-ESO-1 and MAGE proteins exhibit high but often heterogenous expression in melanoma and other epithelial cancers, and are typically absent from somatic tissues with the exception of the cerebellum [12].

Tumours acquire mutations and genetic alterations that can give rise to non-synonymous changes in amino acid sequence, creating peptides that deviate from “self”. These can occur: at the level of single nucleotide substitutions/variations (SNV) that are not corrected by proofreading and repair processes [15]; through insertion or deletion events that induce reading frameshift changes [16,17]; through intron retention or atypical exon-exon splicing events creating novel sequence at junction points [18,19,20]; through aberrant transcription and translation of non-coding regions of DNA [21]; or at the level of large scale chromosomal changes or gene fusions events—one prominent example being the formation of the *BCR-ABL* oncogene in CML and ALL (Figure 1) [22,23]. Several decades of accumulated evidence now suggests that the altered peptide products of these genetic changes, designated “neoantigens”, are among the primary means by which the immune system interacts with a tumour, and that altered peptide repertoires are a vital mediator of both naturally occurring and therapeutic immune-mediated tumour control. This review outlines: the identification and validation of mutated peptide neoantigens as a target of the adaptive immune system in preclinical tumour models and in both retrospective and prospective analyses of patient tumours and tumour-specific T cells; the advances in DNA sequencing and bioinformatic processing technologies that have facilitated rapid and reliable analysis of patient tumour mutational profiles and peptide neoantigen prediction; and the therapeutic modalities by which peptide neoantigens have been, and can be targeted in cancer immunotherapy.

## 2. The Discovery and Characterization of Peptide Neoantigens

The first molecular descriptions of single nucleotide variations inducing immunogenic peptide changes came from a murine model of MCA-induced mutagenesis, with De Plaen et al. demonstrating that a SNV detected in an immunogenic subclone of P815 was the defining element in T cell recognition of that line [24], with Monach et al. subsequently demonstrating the existence of a CD4^+^ T cell response specific to an L49H amino acid substitution in Ribosomal protein L9 after UV-induced mutagenesis [25]. These seminal studies were followed by the first descriptions of mutated peptide neoantigens in melanoma patients within CDK4 [26] and across an aberrantly expressed intron-exon boundary [19].

Concurrently, from the 1980’s onwards, researchers and clinicians at the US National Institutes of Health developed a program of treating metastatic melanoma patients using adoptive transfer of autologous isolated and expanded tumour-infiltrating lymphocytes (TIL), theorizing that ex vivo enrichment and expansion of any pre-existing but in situ-suppressed tumour specific-cells would provide a therapeutic benefit, independent of knowledge of the specifics of their antigen recognition [27,28,29]. Through optimization of patient preconditioning regimens and T cell isolation and expansion strategies, this program achieved remarkable results, culminating in a maximal objective response rate of 72%, 1-year survival of >50% and 5- year survival of 29% in cohorts preconditioned with targeted lymphodepleting chemotherapy and total body irradiation [27,28,29]. Importantly, of those exhibiting complete remissions at the conclusion of the trial, 100% remained alive and in remission at 3 years and 93% at 5 years [28], demonstrating an immune-mediated curative response equivalent to that observed in anti-PD-1 checkpoint blockade therapy [30], which also acts to alleviate in situ suppression of tumour-specific lymphocytes.

Even though these expanded TIL were validated as recognizing autologous tumour cells in vitro at time of administration, the specific nature of the tumour-specific peptides being responded to was not fully characterized. It was known that only a small proportion of responsive cells recognized MDA, and TIL infusion was not associated with the autoimmune side effects observed when MDA were directly targeted [10,27,28,29,31]. Subsequent characterization of archived TIL that had been infused into patients and successfully induced durable remissions revealed dominant TIL reactivity against individual patient neoantigens derived from b-catenin [32], PTPk [33] and p14ARF [34]. Importantly, Lennerz et al. [35] demonstrated that neoantigen-specific reactivity dominated over MDA-specific reactivity when quantified ex vivo. Together these studies provided strong circumstantial evidence that neoantigen-targeted immunity had been directly responsible for remission in these patients.

These seminal findings were achieved through relatively laborious cDNA library synthesis, sequencing and screening, but subsequent technological advances and lowered operational costs in high-throughput next-generation whole genome and whole exome sequencing facilitated a new burst in neoantigen prediction and detection [36]. Initial profiling of human tumours by whole exome [37] and whole genome sequencing [38] confirmed that these approaches could detect high numbers (tens to hundreds) of non-synonymous mutations. Evidence for neoantigen-directed anti-tumour efficacy was first directly demonstrated in a murine B16F10 melanoma model, where Castle et al., utilizing a whole exome sequencing approach, identified 563 expressed neoantigens and undertook immunization using long peptides covering 50 selected candidates. Peptide immunization expanded neoantigen-specific cells and provided tumour control [39]. Subsequently, Matsushita et al. demonstrated that an MCA-treated d42m1 sarcoma line grown in *Rag2*^−/-^ mice, and thus unaffected by any T cell immunoediting during tumourigenesis, could be rejected in wild-type mice on the basis of a T cell response directed against an acquired mutation in spectrin-b2 [40]. This approach was then applied to human patients using cell lines or resected lesions derived from melanoma patients that had exhibited durable responses to TIL or checkpoint blockade therapies, utilizing the first applications of comparative (tumour vs. matched healthy patient tissue) whole exome sequencing [41,42]. These seminal studies confirmed that neoantigen-specific T cells dominated over MDA-responsive T cells in TIL, that both CD4^+^ and CD8^+^ neoantigen-specific responses could be observed, and that individual melanoma patients typically exhibited responses to a small number of patient-specific mutations [41,42,43,44,45]. Expansion of this methodology (further detailed in Section 4.1) has allowed the detection of neoantigen-reactive T cells and characterization of their neoantigen targets in Melanoma (frequently) [28,41,42,43,44,45,46,47,48,49,50], lung cancer, colorectal cancer, squamous cell carcinoma, lymphoma, gastric cancer, ovarian cancer and in some haematological malignancies [20,51,52,53,54,55,56,57,58]—although often in case studies or small cohorts. Taken together, these important proof-of-concept studies have demonstrated the utility of therapeutically targeting mutated peptide neoantigens across a broad range of cancers, and the therapeutic modalities by which this has been addressed are detailed in Section 6.

## 3. Global Analyses of Tumour Mutational Burden

Large, pan-cancer dataset analyses have demonstrated a high degree of variation between (and within) the total tumour mutational burden (TMB) of cancers of different tissue origins [59,60,61]. Some childhood brain and haematological cancers characterized by well-known driver mutations accumulate low numbers of somatic mutations (median 0.01–1 mutation/Mb) while tumours induced by exposure to environmental mutagens (melanoma and lung cancer) typically exhibit >10–100 mutations/Mb. Interestingly, some cancers exhibit detectable signatures of mutation consistent with specific mutagen exposure—for example melanomas are typically characterized by a high frequency of C-T or CC-TT transitions occurring at dipyrimdine sites, consistent with the known effects of ultraviolet radiation; renal cell carcinomas exhibit a high frequency of indel events; while lung adenocarcinomas exhibit a high frequency of CC-AA/AG transversions, associated with the effects of cigarette smoke [59,60,61]. Indeed, exposure of human cell lines in vitro to acetaldehyde, a component of cigarette smoke, recapitulates this mutational signature [16]. While glioma, colorectal cancer and uterine carcinoma typically display low-moderate TMB, subsets of patients within these cancers (and others) exhibit a “hypermutated” phenotype, with median >100 mutations/Mb. This phenotype is typically characterized by loss-of-function mutations or transcriptional silencing in genes encoding important DNA synthesis and mismatch repair proteins, particularly POL*E* and POL*D1* [62,63], MSH1/2, MLH1 [64] and BRCA1/2 [65]. These cancers are classified as being “mismatch repair (MMR) deficient” or in the case of colorectal cancer, displaying “microsatellite instability”. Across all cancers, MMR deficiency is correlated with TMB and TMB, predicted neoantigen load, TIL presence and MMR status all independently correlate positively with response rates to checkpoint blockade with a-PD1 and a-PD-L1 antibodies [30,51,64,66,67,68,69,70,71,72,73]. This association is intuitive as a-PD1/L1 interventions act to alleviate in situ inhibition of existing antigen-specific T cells rather than acting directly on tumour cells [46,74]. Further, melanomas typically exhibit both the highest frequency of neoantigenic mutations and the highest frequency of detectable tumour-specific TIL [27,75]. Although TMB and breadth of neoantigen-specific responses might be assumed to be directly related, there is evidence to suggest that neoantigenic mutations are most likely to elicit a T cell response above a certain threshold of clonality, possibly suggesting that these are “founder” mutations carried by cells emerging early in tumourigenesis and may more efficiently prime T cell responses before the establishment of a fully suppressive tumour microenvironment [76,77]. Interestingly, loss of heterozygosity at *HLA* loci, i.e., the loss of capacity to present certain peptides to T cells has also been shown to be associated with an increase in overall TMB and frequency of neoantigen detection in lung cancer patients, suggesting escape from an ongoing immunoediting process [78].

## 4. Prediction, Identification and Validation of Neoantigens

Despite the high TMB in several cancers, it has been estimated that only 0–5% of potential neoantigenic sequences within any given tumour can generate immunogenic neoantigenic peptides [79]. As peptide neoantigens have become a recognized and efficacious means by which tumours can be targeted, the capacity (and technology) to accurately identify and prioritize immunogenic patient-specific neoantigens has become increasingly important, as has our understanding of the biochemical properties that determine neoantigen processing and presentation efficiency. At present, putative neoantigens can be predicted in silico from comparative healthy and tumour DNA sequencing data, or by direct capture and analysis of MHC-I/II ligands from the tumour cell surface (Figure 2).

### 4.1. In Silico Sequence-Based Neoantigen Prediction

Although no consensus pipeline or systematic guidelines for in silico prediction exist, a general approach is summarized below. An ever-increasing suite of bioinformatic tools are being developed and refined to increase accuracy and efficiency at each step of this process. The list of bioinformatic programs highlighted in-text is by no means exhaustive and is extensively reviewed elsewhere [79,80].

Tumour-specific sequence variations are called by comparison with donor-matched healthy DNA following whole genome or exome sequencing. It is important to note that tumour samples are often heterogeneous and contain varying levels of healthy stroma and immune infiltrate. As such, low frequency sub-clonal SNV may be difficult to call reliably. Further, tumours that exhibit high levels of intratumoural heterogeneity may require multiple pooled biopsies for full clonal coverage [81]. Bioinformatic tools to enhance the detection of subclonal variants or stroma-rich samples include MuTect2 [82] and Strelka [83]. In parallel to the cataloguing of somatic variants, patient *HLA*-haplotype will typically be determined (via Optitype [84] or HLAScan [85]) in order to inform the peptide:MHC (p:MHC)-specific binding parameters that can be applied, and RNA-SEQ data will be collected in order to validate whether variant-containing sequences are actually transcribed and to what level, as some mutated loci have been shown to be silenced and there is evidence that peptide presentation in the context of MHC-I is related to transcript abundance [86].

Key determinants of neoantigen presentation include p:MHC binding affinity and stability. Tools such as SYTHPEITHI [87], NetMHCPan [88], NetMHCStab [89] and MHCFlurry [90] facilitate predictions of binding and stability for most known class I and class II [91] *HLA* alleles, although the strength of data around the key binding determinants is not always equivalent. For example, much of the literature has historically focused on the p:MHC association parameters for HLA-A*02:01 due to the frequency of expression and capacity to present canonical viral epitopes. Typically, less is known about the key determinants of MHC-II binding because of the greater flexibility of the open binding pocket and promiscuity of peptide association.

As a large portion of the MHC-I presented peptide pool derives from breakdown products and defective ribosomal products targeted for ubiquitination [92], the capacity for a neoantigen-containing peptide sequence to be processed and presented is an important consideration. Similarly, within the MHC-II processing and presentation pathway sensitivity to serine, aspartic and cysteine proteases is important for peptide trimming and secondary peptide structure is predictive for cleavage sensitivity, as MHC-II epitopes are not full denatured before trimming [93,94]. As such, additional filters that assess proteasomal processing (NetChop [95]) and enzymatic processing have been developed (PapRoc for MHC-I [96] and PepCleaveCD4 for MHC-II [93]). Pipelines that incorporate as many of these filters as possible tend to give the most accurate predictions as to which putative neoantigens will prove to be immunogenic—for instance Tang et al. demonstrated that a TruNeo pipeline that fully incorporates and differentially weights: MHC-I binding affinity; proteasomal C-terminal cleavage; TAP transport efficiency; expression abundance by RNA-SEQ; clonal heterogeneity and HLA-allele-specific loss of heterozygosity (to avoid false positives from neoantigens restricted by non-tumour-expressed MHC) outperformed analyses based on MHC-binding affinity-based algorithms alone (NetMHCPan and MHCFlurry) in predictive power and accuracy in an analysis of sequence from a lung cancer patient and in retrospective analyses of published datasets [97].

### 4.2. Neoantigen Detection by Mass Spectrometry

As an alternative to, or in synergy with, in silico sequence-based predictions, liquid chromatography-mass spectrometry (LC-MS) can be used to directly interrogate the “immunopeptidome” or “ligandome” of tumour cells. One of the initial limitations of this technology was the requirement for a large amount of starting material (~10^8^ cells or 1 g of tumour sample), necessitating a reliance on autologous cell lines that risked in vitro deviation from in vivo patterns of presentation [98]. Recently, refinements to LC-MS have allowed ligandome detection from 0.1g of tumour material [99,100]. Typically, p:MHC complexes are captured from cell or tumour lysates by column immunoprecipitation, then peptides eluted by acidification for LC-MS. In contrast with traditional total proteomic analyses, where proteins are tryptically digested, leaving all peptides with a basic C-terminal residue, eluted peptides are not enzymatically processed and are biochemically diverse. This can introduce biases in LC tractability—as such, uniform di-methylation of all amino acid side chains increases peptide hydrophobicity and can increase peptide detection depth by a factor of two [99,101]. Typically, putative ligandome candidates are far more numerous than those detected by initial in silico filtering, as the whole of the normally presented peptidome is also captured. As such, detection by LC-MS can be somewhat inefficient and laborious—in one recent example Bassani-Sternberg et al. identified 95,000 eluted peptides, of which only 11 were candidate mutated sequences and 4 proved to be immunogenic [100]. This study was also able to characterize expression of peptides derived from MDA and CGA, but expression of these is typically also captured by RNA-SEQ in an in silico approach. Candidate neoantigens detected by LC-MS can be compared to matched in silico predictions, strengthening the predictive power of each [102] and, importantly, ligandome data provides information about the minimal presented peptide epitope incorporating an amino acid substitution, eliminating the need for extensive epitope mapping. Additionally, specialized tools for ligandome peptide mapping such as SpectMHC [103] have been developed. Interestingly, some epitopes of low predicted MHC binding affinity, notably the R175H neoantigen derived from TP53, have been captured by LC-MS ligandomics—acting as an important confirmation of their genuine presentation and potentially suggesting that LC-MS based approaches may be less likely to exclude candidate neoantigens through false negative errors. LC-MS approaches have also revealed that the nature of the tumour ligandome can be influenced by immune activity in the tumour microenvironment—Wickstrom et al. found that ligandome-validated peptides from resected tumour were only expressed on patient-matched tumour cell lines in vitro after treatment with IFN-g [104]. IFN-g has well-characterized effects on modulating the expression of peptide processing machinery, increasing expression of ERAP-1 and biasing away from expression of the normal b2 proteasome subunit towards expression of subunits b1and b5, typically components of the “immunoproteasome” primarily expressed in APCs [105,106,107,108]. As such, the ligandome of a “hot” intratumoural environment replete with active T cells producing IFN-g may not be perfectly reflected in vitro by autologous cell lines, an important consideration for neoantigen validation (further discussed in Section 4.4).

### 4.3. Important Correlates of Neoantigen Presentation

Even among pipeline-validated neoantigens, a large proportion do not elicit T cell responses. Several studies have investigated the characteristics that help to ultimately determine neoantigen immunogenicity. Typically, MHC-I restricted neoantigens can be classified as exhibiting an amino acid change at an anchor residue (positions 2 and 8–9) or at a non-anchor core residue (positions 3–7). As p:MHC anchorage is a key determinant of MHC-I binding affinity, non-anchor neoantigens typically exhibit equivalent p:MHC affinity to their equivalently presented wild-type comparators [109]. Due to this, it is likely that many T cells bearing potentially reactive TCR will have been deleted during negative selection in the thymus. As such, in these cases several groups have demonstrated that the degree of biochemical difference in amino acid substitution is a key determinant of immunogenicity, particularly at positions 4–6 where amino acid side chains interact with the TCR CDR3 loop. Calis et al. showed that mutations leading to the incorporation of amino acids with large aromatic side chains (for example phenylalanine) were more frequently immunogenic [110]. Further, Capietto et al. and Yadav et al. demonstrated, using candidate neoantigens predicted from MC38, TRAMP-1, EMT6 and CT26 murine tumour lines, that non-anchor amino acid changes that increased p:MHC stability and therefore duration of TCR contact also increased immunogenicity [15,111].

By contrast, in anchor residue mutations, the TCR interface remains equivalent between neoantigenic and wild-type peptide sequence, and for this class of neoantigen the key determinant in immunogenicity is the degree of difference in MHC-I binding affinity [112,113]. As wild-type peptides lacking canonical anchor residues will bind poorly to MHC-I, these are less likely to be presented on MTEC and induce deletion of T cells bearing TCR capable of interacting with the shared peptide core sequence.

### 4.4. In Vitro Validation Assays

A lack of patient TIL response to a putative neoantigen does not directly indicate that the neoantigen is not immunogenic. TIL are highly heterogenous, typically comprising large numbers of virus-specific T cells and other effectors migrating into tumour sites along the same chemokine gradients as neoantigen and CGA-specific cells [114,115,116]. Further, pre-existing neoantigen-specific populations can be impacted by chemotherapeutic treatment. However, detection of such a response, especially when confirmed by TIL reactivity to autologous tumour cells (in the presence absence of IFN-g, as discussed in Section 4.2) is the clearest possible indication that a putative neoantigen is immunogenic.

In order to validate T cell recognition of putative neoantigens, candidates are synthesized into long peptide sequences (~20–25mer) that capture all possible peptide epitope ‘frames’ for MHC-I/II by positioning the mutated residue centrally; or are encoded as tandem minigenes in viral vectors [117,118]. These can be loaded or transduced, respectively, into autologous donor APC and co-cultured with TIL or PBMC. The tandem minigene approach has the added advantage of demonstrating that an expressed neoantigen is processed and presented by endogenous MHC-I presentation machinery. T cell responses can be detected by proliferation, cytokine production or activation marker expression (for instance 4-1BB or CD25) [119,120] and overlapping short peptide pools can be used to determine minimal peptide epitopes (MPE) required for T cell activation in order to completely to define the neoantigenic peptide epitope. Where MPE are already available these can be incorporated into p:MHC multimers, and T cell specificity for up to 20, ~100 or ~1000 peptides can be simultaneously assessed by labelling of multimeric constructs with differential fluorophores (flow cytometry), isotopes (mass cytometry) or DNA barcodes, respectively [43,116,121]. Further, longitudinal analysis of clonotype expansion after additional therapeutic interventions (for example a-PD-1 or peptide vaccination) can be tracked using either p:MHC multimers or TCRVb deep sequencing [122].

## 5. Public Neoantigens

The vast majority of detected neoantigens are patient-specific, or “private”, necessitating a personalized approach to adoptive cell therapy. As mutations are acquired somewhat stochastically, private neoantigens occur more frequently (although not exclusively) in loci non-essential for tumourigenesis and metastasis, termed “passenger” mutations. By contrast, mutations in some important driver oncogenes have been shown to occur in “hotspots” and be shared across multiple patients [123]. Where patients express both a shared mutation and a shared MHC molecule capable of presenting the encoded neoantigen in an immunogenic context, this pairing can be designated as a “public” neoantigen. Characterized public neoantigens generally occur in genes that play an important role in facilitating tumourigenesis or driving continued tumour growth, for instance the gain-of function BRAF V600E mutation that confers a constitutive proliferative drive in melanoma [124]. Public neoantigens are most frequently described in driver oncogenes. The prominent exception is *TP53*, a master orchestrator of cell cycle and DNA repair processes. *TP53* is the most frequently mutated gene among all cancers, with *TP53* mutations represented across at least 27 cancer types [125]. As such, these targets may be more difficult for cancer cells to lose or silence without a concomitant loss of fitness.

Public neoantigens and their shared MHC restrictions have been described in several cancer types, frequently with overlapping MHC-I and MHC-II epitopes. This phenomenon of epitope “nesting” has also been described in CGA (for example in the overlapping HLA-A*0201-restricted ESO_157-165_ and HLA-DPB1*04-restricted ESO_157-170_ peptides) and may be an important means of inducing a concurrent CD4^+^ and CD8^+^ T cell response [13]. A non-exhaustive list (reviewed in detail in Pearlman et al. [123]) of public neoantigens within oncogenic drivers for which TIL reactivity has been demonstrated include: CDK4 R24C restricted by HLA-A*02 in melanoma [31]; BCR/ABL b3a2 junction restricted by HLA-A*03 in CML [23]; BRAF V600E restricted by HLA-A*02:01, -B*27:05, HLA-DRB1*04 and -DQB1*03 in melanoma [50,114,126], and KRAS G12C/D/V restricted by HLA-C*08 and HLA-DRB1*07 in pancreatic, colorectal and endometrial cancers [127,128,129,130]. An array of public neoantigens exists within a mutational hot spot encoding amino acids 175–282 within TP53, including: R248W (HLA-A*02; HLA-DRB1*13:01 in colorectal cancer); R245S (HLA-DRB3*0202 in ovarian cancer); Y220C (HLA-A*02; HLA-DRB1*04 in colorectal cancer and HLA-DRB3*02:02 in ovarian cancer) and R175H (HLA-A*02:01 and HLA-DRB1*13:01 in colorectal cancer) [118,120,131,132,133]. *TP53* hotspot mutations and amino acid changes typically impact the DNA-binding domain and disrupt the ability of *TP53* to sense DNA damage and orchestrate the Mre11/ATM-dependent DNA damage response [134].

The utility of targeting a public neoantigen (above and beyond individualized patient treatment) can be determined by both the frequency of the mutation/amino acid substitution and the frequency of expression of the restricting *HLA* allele in a given population. As an example, one recent review estimates that KRAS G12D is targetable in ~10% of all pancreatic cancer patients, and that across the most common cancer types in the USA (colorectal, renal, lung, endometrial and cervical) G12D mutant peptides presented in the context of HLA-A*03 could be targetable in ~1% of all patients, numerically equivalent to many specific small molecule inhibitor therapies [123]. Further, as public neoantigens are more likely to be clonally expressed than patient-specific neoantigens and as they typically effect proteins essential for tumour fitness, antigen loss tumour escape variants may be less likely to arise. Importantly, both public and private neoantigens have been reported to occur within the same tumours. Public neoantigens can be targeted using the same therapeutic modalities as private neoantigens, discussed below.

## 6. Therapeutic Targeting of Neoantigens: The Past, Present and Future

### 6.1. TIL Therapy

As discussed in Section 2, the remarkable success in utilizing adoptive transfer of expanded TIL to treat metastatic melanoma has been largely attributed to neoantigen-specific responses based on retrospective analyses of archived TIL fractions [42,43,47]. As an example, deep TCRVb sequencing of 12 melanoma TIL revealed that in each patient up to five of the most dominant TCRVb clonotypes were specific for neoantigens rather than MDA or CGA [122]. As tumour-reactive cells typically only comprise a small fraction of all TIL, TIL therapy is mediated by culture of pooled TIL with autologous tumour cells, with those TIL sub-pools that exhibit anti-tumour responsiveness (for instance via the production of IFN-g) allowed to briefly expand in situ (to enrich for antigen-specificity) then polyclonally re-expanded (for instance through stimulation using OKT3 and interleukin-2) [27,135]. This culture methodology frequently allows 10^9^-10^11^ T cells to be adoptively transferred. Recent targeted prospective studies characterizing neoantigen-specificities in TIL (through the pipelines described in Section 4) prior to infusion have similarly shown strong anti-tumour efficacy. In one landmark study, Tran et al. identified an ERRB2IP-derived mutation in a metastatic cholangiocarcinoma patient. Infusion of TIL enriched for CD4^+^ T cells recognizing this neoantigen induced disease stabilization and regression of discrete lesions. Subsequently, additional therapy using a pure population of ERRB2IP-neoantigen responsive CD4^+^ T cells induced a complete and durable remission [136]. Similarly, infusion of expanded TIL comprising four clonotypes directed against the public neoantigen KRAS G12D in a colorectal cancer patient induced the complete disappearance of 6 of 7 detectable lung metastases—with one lesion initially regressing then outgrowing after losing detectable expression of the restricting HLA-C*08:02 class I molecule [137]. Infusion of expanded TIL targeting four neoantigens across HLA-B, -C and -DRB1 restrictions induced a complete response in a breast cancer patient refractory to chemotherapy, ongoing for >22 months [55]. Finally, Comoli et al. have reported that priming, expansion, and adoptive transfer of BCR-ABL junction site-specific T cells from patients and *HLA*-matched healthy donors induced complete remission [20]. Taken together these studies indicate that prospective targeting of neoantigens utilizing expanded TIL remains an efficacious therapeutic approach.

### 6.2. TgTCR, CAR-T and Antibody-Mediated Therapies

Retrospective analyses of the phenotypic characteristics of infused TIL in melanoma patients suggested that retention of telomere length, co-stimulatory marker expression and a detectable central-memory subset consistent with lesser differentiation status were important correlates of clinical efficacy [27,28]. As serial stimulation of TIL runs the risk of terminal differentiation prior to adoptive transfer, strategies to confer neoantigen reactivity to defined naive and memory T cell populations derived from patient PBMC have been established, allowing for greater control over T cell culture phenotype and function. Several studies utilizing transgenic (Tg)TCR murine models (for example targeting epitopes derived from gp100/pmel) have demonstrated that effectors derived from naive, central memory or stem-like memory precursors provide optimal anti-tumour efficacy [138,139,140]. TCR sequences can be readily captured from neoantigen-specific T cells, following in silico neoantigen prediction and validation, and transduced or transferred into autologous T cells through lentiviral or retroviral vectors [50,57], CRISPR editing [141] or by use of ‘sleeping beauty’ transposon/transposase systems [142]. Each of these facilitates the silencing of endogenous TCRa/b chains to avoid mispairing that may create autoreactive TCR. This strategy has recently been used to target the CGA NY-ESO-1 in melanoma, with notable clinical efficacy [143,144]. The use of neoantigen-targeted TgTCR against pipeline-derived candidate neoantigens has been used in preclinical models in acute myeloid leukemia and is rapidly progressing into the clinic. Clinical trials utilizing TgTCR targeting KRAS G12V (NCT03190941) and G12D (NCT03745326) in the context of HLA-A*11:01 and utilizing up to five personalized TgTCR (NCT03412877) are currently being carried out across a wide range of cancer indications [145]. Importantly, the development of an extensive library of constructs encoding TgTCR for validated public neoantigen/MHC pairings will continually broaden the practicality and applicability of this therapeutic strategy. Further, the use of these validated TCR in bispecific “immune-mobilizing-monoclonal-TCR-against-cancer” (ImmTac) formats, whereby the TCR is coupled to a stimulatory a-CD3 antibody to bind and activate any local T cells to cognate p:MHC expressing targets are also in development [146].

Antibodies, or single-chain-variable fragments (scFv) that recognize specific p:MHC pairings as their cognate epitope (described as “TCR mimics”) have been isolated, often through phage display screening. The reformatting of scFv as single chain diabodies directed against TP53 R175H [147,148] and KRAS G12D [147] in a bispecific format with a-CD3 to activate local T cells in a similar manner to ImmTac has been reported. Finally, chimeric-antigen-receptor (CAR) T cells utilize a scFv coupled through a transmembrane domain to signaling domains from CD3z, CD28 and/or 4-1BB. Most prominently deployed against CD19-expressing cells in haematological malignancies, where they have achieved remarkable clinical efficacy [149], the only current clinically relevant mutant antigen target for CAR-T cells is the exon 2-7 deletion vIII variant of EGFR, commonly expressed in glioma [150,151]. CAR-T incorporating a TCR mimic scFv specific for the HLA-A*02:01-retricted NY-ESO-1_157-165_ epitope as their antigen binding domain have been reported [152], reinforcing the tractability of this approach and CAR-T directed against public neoantigens may represent a promising future therapeutic modality.

### 6.3. Neoantigen Vaccination Strategies

Candidate neoantigens detected by in silico or LC-MS pipelines can be formulated into vaccines designed to restimulate existing responses in patient TIL and PBMC, or to induce the de novo priming of new T cell responses absent in pre-existing TIL. This approach has the theoretical advantage of not requiring the lengthy and costly in vitro expansion of a T cell product for adoptive transfer. The efficacy of neoantigenic vaccines was first demonstrated in murine models where 50 candidate neoantigens detected from B16F10 melanoma elicited tumour control when administered as a synthetic long peptide pool [153], and candidate neoantigens from 4T1 mammary carcinoma and CT26 colorectal carcinoma lines elicited tumour rejection when formulated as an mRNA vaccine [39]. In both approaches CD4^+^ T cell responses were initially predominant, with subsequent epitope spreading facilitating expansion of neoantigen-responsive CD8^+^ T cells.

Several vaccine studies have targeted brain cancers, typically poorly responsive to checkpoint blockade and other immunotherapeutic modalities. Personalized candidate neoantigen peptide pools have elicited T cell responses in glioma and astrocytoma, with some evidence of disease stability [154,155,156]. Schumacher et al. administered a synthetic long peptide vaccine against the public neoantigen R132H in IDH1, after validation in *HLA*-humanized mice where it elicited potent tumour control of R132H mutant-tumours. This study reported CD4^+^ T cell responses in 4 of 25 patients across a broad HLA-DR repertoire [157].

Autologous dendritic cells (DC) loaded with oxidized whole-tumour lysate have been shown to elicit neoantigen-specific responses in an unbiased fashion in ovarian cancer—promoting the expansion of neoantigen-responsive cells detected in the blood of patients and priming previously undetected responses to in silico predicted neoantigens [158]. In a more targeted study, Carreno et al. loaded autologous DC with 10 prioritized neoantigens paired to extremely low frequency preexisting responses in melanoma patient PBMC. Each of these was prominently expanded by vaccination, and interestingly, although neoantigen-specific responses within each patient were limited to individual epitopes, TCR clonal diversity against these increased, suggesting priming of new clonotypes [159].

Two vaccination studies of particular importance have demonstrated clinical efficacy in stage III-IV melanoma patients. Sahin et al. demonstrated a significant slowing of disease progression after administration of prioritized candidate neoantigens as synthetic mRNA, with objective responses in two of five patients with stage IV disease. All patients enrolled in this trial exhibited responses to at least 3 predicted neoantigens, and neoantigen-specific responses were far greater in magnitude than those against co-administered CGA and MDA sequences [160]. Similarly, Ott et al. administered synthetic long peptides covering 20 prioritized candidate neoantigens adjuvanted by co-administration of the TLR agonist POLY:IC-LC. Four of six patients exhibited no disease progression over 25 months, and two stage IV patients that did progress underwent durable regression following a-PD1 therapy, emphasizing the synergy between vaccination restimulating and/or inducing neoantigen-specific responses, and checkpoint blockade allowing the expanded cells to retain intratumoural effector function. All patients exhibited both CD4^+^ and CD8^+^ responses and longitudinal tracking showed that these cells persisted and that the neoantigen-specific repertoire broadened over time, suggesting a profound repolarization of the tumour microenvironment [161].

### 6.4. Synergies with Checkpoint Blockade

TMB, TIL frequency, MMR deficiency and predicted neoantigen load all correlate with clinical responses to checkpoint blockade by both a-CTLA4 and a-PD-1 across a range of cancers [30,46,51,64,67,68,69,70,71,72,162,163]. Checkpoint blockade targeting the PD-1/PD-L1 axis exerts its immunomodulatory function by alleviating in situ suppression of existing tumour-reactive T cells, as PD-1 signaling impairs both TCR/CD3- and CD28-mediated stimulatory and co-stimulatory signal integration, respectively [164,165]. Neoantigen-specific T cells typically express PD-1 in vivo, and Gros et al. and others have shown that selection of PD-1^+^ and/or CD39^+^ lymphocytes from within TIL or from the peripheral circulation enriches for neoantigen-reactive cells [119,122,129,166]. Interestingly, several studies have demonstrated that in both chronic viral disease and in established tumours (i.e., settings where T cells can become functionally ‘exhausted’) the antigen-specific cells that undergo a profound proliferative burst following a-PD-1 administration exhibit a defined PD-1^+^ TCF-1^+^ phenotype, demarcating them as “precursor-exhausted” T cells [165,167] (T_PEX_), while more terminally differentiated and fully exhausted PD-1^HI^ Tim-3^+^ TCF-1^−^ cells do not exert clinically relevant anti-tumour function [168,169,170,171,172]. The frequently demonstrated capacity of neoantigen-specific T cells within TIL to expand and exhibit polyfunctional cytokine production ex vivo suggests that these cells exist within an intratumoural niche as T_PEX_. Indeed, evidence from a recent study utilizing a murine Lewis lung carcinoma model demonstrated that neoantigen-reactive cells within TIL exhibited a T_PEX_-like PD-1^+(DIM)^ SLAMF6^+^ phenotype [173], mirroring that described in human tumours. Further, in melanoma and HNSCC notable expansion of neoantigen-specific T cell populations has been observed following both a-CTLA4 and a-PD-1 administration, with the expansion of these populations temporally aligning with lesion regression [47,54]. As such, checkpoint blockade is likely to synergize with both cell-mediated and vaccine approaches to stimulate (and maintain) neoantigen-specific immune responses.

## 7. Conclusions

Peptide neoantigens represent an extremely promising, and now well-validated immunotherapeutic target, and offer some realization of a decades-long movement towards precision personalized medicine in the treatment of cancer. Ongoing refinement of tools for optimized detection and prioritization of neoantigens, the development of novel modalities targeting p:MHC, optimization of vaccine delivery and adjuvant formulation and refinement of techniques in the culture of adoptive cell therapy products will continue to advance the implementation of these therapies. As neoantigen-specific TIL are typically inhibited in situ by other factors beyond the PD-1/PD-L1 axis, a more thorough understanding of both the phenotypic nature of TIL and the stimulatory and inhibitory interactions they undergo within the tumour microenvironment will inform future therapies targeted at maximizing their anti-tumour function.

## Figures and Tables

**Figure 1 cancers-13-04245-f001:**
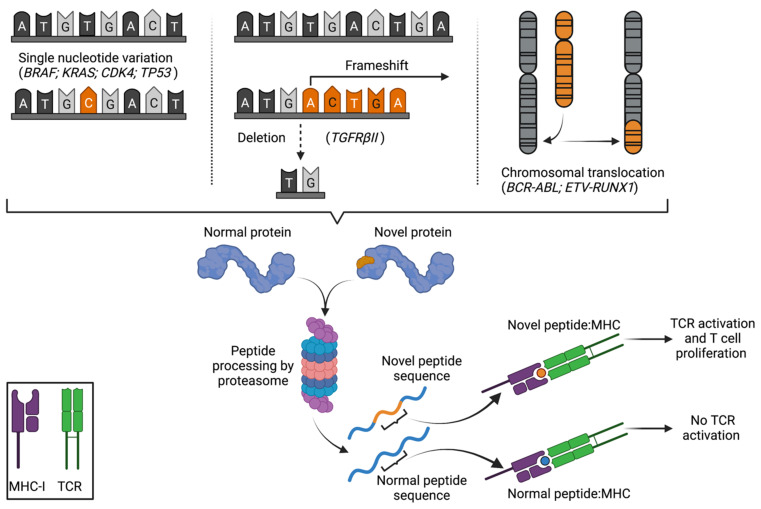
Generation and recognition of neoantigenic peptides after mutational or structural changes to somatic DNA. Changes to coding nucleotide sequence can be generated by non-synonymous point mutations, insertion/deletion events leading to reading frameshifts, or larger-scale structural changes such as chromosomal translocation and gene fusion events. When these changes to somatic DNA cause an alteration in amino acid sequence creating a peptide that can be processed and presented in the context of MHC-I/II and induce TCR activation, such a peptide is designated a neoantigen.

**Figure 2 cancers-13-04245-f002:**
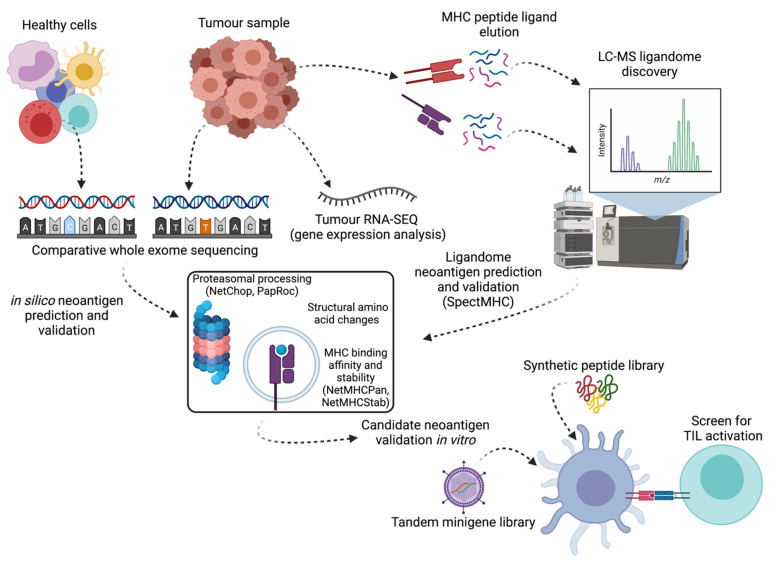
Overview of neoantigen identification and validation pipelines. Matched whole-genome or whole-exome sequencing data is used to identify sequence differences between healthy and tumour-derived DNA. Concurrently, the eluted tumour MHC-I/II ligandome can be assessed via LC-MS, and tumour RNA expression data is collected to validate the presence and expression level of mutated transcripts. Resulting putative neoantigenic peptide sequences are triaged and prioritized using bioinformatic tools to inform likely proteasomal processing, degree of difference from matched wild-type peptide, and MHC binding affinity and stability based on donor *HLA* haplotype. Finally, selected candidates are expressed as tandem minigene libraries or synthesized as long peptides and the capacity of patient T cells to recognize transduced or peptide-loaded autologous APC and autologous tumour cells is assessed. Algorithms listed are cited in-text.

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
