# Peer review of "Identification and Targeting of Mutant Peptide Neoantigens in Cancer Immunotherapy"

_cancers, 2021, doi:10.3390/cancers13164245_

Round 1

Reviewer 1 Report

Overall, this manuscript is written very well, impressing practicability of neoantigen-targeted therapeutics.  However, in reality, neoantigen-targeted therapy is still underway of development and this side should also be discussed.  For example, how long does it take to complete the pipeline of predicting and validating neoantigen peptides and is it acceptable for practical clinical use?  

Small points for correction:

Some abbreviations should have full spelling (for example, CPB.)

Figure numbers are missing from figures.

Author Response

Point 1: Overall, this manuscript is written very well, impressing practicability of neoantigen-targeted therapeutics.  However, in reality, neoantigen-targeted therapy is still underway of development and this side should also be discussed.  For example, how long does it take to complete the pipeline of predicting and validating neoantigen peptides and is it acceptable for practical clinical use?  

Response: The authors agree that this is an important consideration and did attempt to make clear that neoantigen prediction pipelines and therapies are still typically undertaken in small exploratory cohorts and do not currently constitute standard clinical practice.  The authors have further highlighted in-text that this field is moving towards greater utilisation in the clinic through the initiation of several larger scale clinical trials. It is difficult to make absolute statements about the length of time taken to complete a pipeline of neoantigen prediction and validation, as this will vary widely based on availability of sequencing and bioinformatic analysis, and capacity to undertake in vitro validation and expand GMP-grade adoptive cell therapy products across centres.  However, the review does describe, in detail, several case studies whereby prospective identification of targetable neoantigens has led to effective TIL or vaccination treatments being deployed, highlighting the tractability of this approach clinically.  The authors have also included the statement in section 6.3 that vaccination approaches have; “the theoretical advantage of not requiring the lengthy and costly in vitro expansion of a T cell product for adoptive transfer”, recognising that GMP-grade expansion for adoptive cell therapy products is a potential roadblock to the widespread utilisation of these technologies.

Point 2: Small points for correction:

Some abbreviations should have full spelling (for example, CPB.)

Response: We thank the reviewer for highlighting this oversight, which has now been corrected. All nomenclature that might be abbreviated is written in full for its first appearance in text, followed by abbreviation in brackets and in the abbreviations list at the end of the review. All instances of CPB have been changed to ‘checkpoint blockade’ in full.

Point 3: Figure numbers are missing from figures.

Response: We thank the reviewer for highlighting this issue. It appears that the figure titles and legends were not properly incorporated into the version sent out for review, but this has now been addressed.  

Reviewer 2 Report

The review summarized the literature about mutated peptide neoantigens for the treatment of cancer. It is a good description of the literature, although some aspects should be more clearly described.

Some aspects of it must be addressed before the publication.

First of all the article needs on the first page a simple summary to briefly describe the review.

Secondly the definition of "public and private" term are difficult and they should reconsider simpler descriptions for these terms (in the title and in the abstract).

Thirdly, a table should be made to describe some of the neoantigens strategies that are currently available.

In the last part of the introduction where the authors mention extensive evidence they should also make extensive references.

In the second half of the second section the manners of production and detection should have adequate references as well as it should have some informatics methods described.

At the end of the second section they must explain small ongoing clinical studies. Explain more the results and what is the field going through at the moment.

At the end of the third section it must be mentioned that progress needs to be made to find best biomarkers as predictors as not all tumors have an MMR status.

In the picture put some examples of algorithms for Neoantigens detection as LCMS. Put the other in silico sequence-based neoantigen prediction methods in it.

I would consider removing section 5 about public neoantigens definition.

At the end of section 6.2 there is a typo : <<may represent “an” promising future>>

Author Response

The review summarized the literature about mutated peptide neoantigens for the treatment of cancer. It is a good description of the literature, although some aspects should be more clearly described.

Point 1. First of all the article needs on the first page a simple summary to briefly describe the review.

Response: We thank the reviewer for pointing out this oversight – the simple summary has now been added to the manuscript as detailed below:

Simple Summary: Cancerous cells acquire genetic mutations that can lead to changes in the amino acid sequence of proteins. These altered amino acid sequences, or “neoantigens” allow the immune system to recognize the mutated cells as “non-self” and attack them. This review details the discoveries that identified neoantigens as a key target of the immune system, the development of bioinformatic and DNA sequencing technologies to detect patient-specific mutations that can give rise to neoantigens, and the methods by which neoantigens can be targeted in cancer therapy.

Point 2: Secondly the definition of "public and private" term are difficult and they should reconsider simpler descriptions for these terms (in the title and in the abstract).

Response: We thank the reviewer for sharing this concern.  In the interests of clarity, we have removed “public” and “private” form the review title and have adjusted the abstract to read: “prioritising both patient-specific or “private” and frequently occurring, shared “public” neoantigenic targets.  However, as the terms “public” and “private” are well-established in the literature and universally used, therefore the authors do not believe it is appropriate to deviate from their use. 

Point 3: Thirdly, a table should be made to describe some of the neoantigens strategies that are currently available.

Response: We agree that in general, tables are an excellent means of summarising detailed data that may not be fully engaged with in-text. However, in this case as each 1) method of neoantigen detection and validation and 2) therapeutic strategy utilised to target neoantigens is given its own detailed section of discussion, the authors feel that repetition of this information in table format would be somewhat repetitive and would not necessarily add value to the review.   

Point 4: In the last part of the introduction where the authors mention extensive evidence they should also make extensive references.

Response: We have added the following text: “Several decades of accumulated evidence now suggests that the altered peptide products of these genetic changes, designated “neoantigens”, are among the primary means by which the immune system interacts with a tumour, and that altered peptide repertoires are a vital mediator of both naturally occurring and therapeutic immune-mediated tumour control. This review outlines: the identification and validation of mutated peptide neoantigens as a target of the adaptive immune system in preclinical tumour models and in both retrospective and prospective analyses of patient tumours and tumour-specific T cells; the advances in DNA sequencing and bioinformatic processing technologies that have facilitated rapid and reliable analysis of patient tumour mutational profiles and peptide neoantigen prediction; and the therapeutic modalities by which peptide neo-antigens have been, and can be, targeted in cancer immunotherapy”.      

Point 5: In the second half of the second section the manners of production and detection should have adequate references as well as it should have some informatics methods described.

Response: We authors thank the reviewer for suggesting a lack of clarity in this section, although the authors do consider that it has been heavily and appropriately referenced. Additional references describing early whole-genome and whole exome sequencing studies have been incorporated (Stratton et al 2011, Sjoblom et al 2006 and Pleasance et al 2010), as has a mention of the progressively lowering associated costs.  The authors have made clear that the Castle et al paper detailed utilized a whole exome sequencing approach.  Bioinformatic methods are more completely detailed in section 4, although the technological specifics of the sequencing technologies are well beyond the scope of this review.

Point 6: At the end of the second section they must explain small ongoing clinical studies. Explain more the results and what is the field going through at the moment.

Response: In the interest of transparency, the authors wanted to make explicit the fact, outside of melanoma, that many neoantigen discovery trials have been in relatively small patient cohorts of 1-20, although within these cohorts the frequency of detection of immune responses to neoantigens has been high.  To provide a link between the closing statement of the introduction and the current state of the field, we have added the following text:  “Taken together, these important proof-of-concept studies have demonstrated the utility of therapeutically targeting mutated peptide neoantigens across a broad range of cancers, and the therapeutic modalities by which this has been addressed are detailed in section 6.”   

Point 7: At the end of the third section it must be mentioned that progress needs to be made to find best biomarkers as predictors as not all tumors have an MMR status.

Response: We agree that the search for additional predictive biomarkers of responsiveness to ACT and CPB is a priority in the field.  It is certainly true that only a subset of tumors are MMR-deficient, and that these have a unique capacity to stimulate TIL and respond to CPB due to their higher than normal mutational burden, as explicitly stated in-text.  Although the search for predictive biomarkers of response beyond those listed is an area of very active investigation in the literature, it is beyond the scope of this review.

We do state in-text that: “Across all cancers, MMR deficiency is correlated with TMB, and TMB, predicted neoantigen load, TIL presence and MMR status all independently correlate positively with response rates to checkpoint blockade with a-PD1 and a-PD-L1 antibodies” indicating that irrespective of MMR status TMB, neoantigen load and the presence of TIL are themselves important biomarkers of CPB and ACT response.  The authors have also amended the text to state that melanoma typically harbours both the highest frequency of neoantigens and the highest frequency of tumour-infiltrating lymphocytes, and have additionally cited the work of Galon et al regarding the relationship between intra- and peri-tumoral TIL frequency and patient prognosis.  

Point 8: In the picture put some examples of algorithms for Neoantigens detection as LCMS. Put the other in silico sequence-based neoantigen prediction methods in it.

Response: The authors thank the reviewer for this suggestion, and have amended figure 2 to include specific bioinformatic programs at each step of neoantigen prediction, matching those listed in sections 4.1-4.3 in-text.   We have included publication licenses for each figure and amended figure legends.

Point 9: I would consider removing section 5 about public neoantigens definition.

Response: The authors thank the reviewer for this organisational suggestion.  However, we believe that this section introducing and highlighting the differences between patient-specific (private) and frequently occurring/shared (public) neoantigens is important, especially as the therapeutic modalities targeting them overlap. Further, this section offers a detailed breakdown of which described public neoantigens have actually been validated as being responded to by patient TIL, rather than simply being detected by sequencing pipelines. Finally, prior literature does not always make clear the point that unlike small molecules (for instance vemurafenib targeting V600E), immunotherapeutic targeting of shared neoantigens also requires shared HLA expression. As such, we would argue that introducing and detailing the distinction between public and private neoantigens in a discrete section, rather than during a discussion of therapeutic modalities, adds clarity and value to the review.  

Point 10: At the end of section 6.2 there is a typo : <<may represent “an” promising future>>

Response: This been corrected. 
